# Scaling up Trustless DNN Inference with Zero-Knowledge Proofs

## Abstract

As ML models have increased in capabilities and accuracy, so has the complexity of their deployments. Increasingly, ML model consumers are turning to service providers to serve the ML models in the ML-as-a-service (MLaaS) paradigm. As MLaaS proliferates, a critical requirement emerges: how can model consumers verify that the correct predictions were served in the face of malicious, lazy, or buggy service providers?

We present the first practical ImageNet-scale method to verify ML model inference non-interactively, i.e., after the inference has been done. To do so, we leverage recent developments in ZK-SNARKs (zero-knowledge succinct non-interactive argument of knowledge), a form of zero-knowledge proofs. ZK-SNARKs allows us to verify ML model execution non-interactively and with *only* standard cryptographic hardness assumptions. We provide the first ZK-SNARK proof of valid inference for a full-resolution ImageNet model, achieving 79% top-5 accuracy, with verification taking as little as one second. We further use these ZK-SNARKs to design protocols to verify ML model execution in a variety of scenarios, including verifying MLaaS predictions, verifying MLaaS model accuracy, and using ML models for trustless retrieval. Together, our results show that ZK-SNARKs have the promise to make verified ML model inference practical.

## 1    Introduction

ML models have been increasing in capability and accuracy. In tandem, the complexity of ML deployments has also been exploding. As a result, many consumers of ML models now outsource the training and inference of ML models to service providers, which is typically called "ML-as-a-service" (MLaaS). MLaaS providers are proliferating, from major cloud vendors (e.g., Amazon, Google, Microsoft, OpenAI) to startups (e.g., NLPCloud, BigML).

A critical requirement emerges as MLaaS providers become more prevalent: how can the model consumer (MC) verify that the model provider (MP) has correctly served predictions? In particular, these MPs execute model inference in untrusted environments from the perspective of the MC. In the untrusted setting, these MPs may be lazy (i.e., serve random predictions), dishonest (i.e., serve malicious predictions), or inadvertently serve incorrect predictions (e.g., through bugs in serving code). The models the MP serves may come from the MP itself (in which MP wishes to hide the weights), a third party, or the MC itself. Even in the latter two cases, MC may still be interested in trustless verification if the inputs are hidden.

One emerging cryptographic primitive that could address the problem of verified ML inference in the untrusted setting is ZK-SNARKs (Zero-Knowledge Succinct Non-Interactive Argument of Knowledge). ZK-SNARKs are a cryptographic primitive in which a party can provide a certificate of the execution of a computation such that no information about the inputs or intermediate steps of the computation is revealed to other parties. Unfortunately, there are no practical solutions for realistic model sizes, such as ImageNet-scale models. Prior work on ZK-SNARKs for ML models is limited to toy datasets such as MNIST or CIFAR-10 Feng et al. (2021); Weng et al. (2022); Lee et al. (2020); Liu et al. (2021).

In this work, we present the first ZK-SNARK circuits that can verify inference for ImageNet-scale models. We are able to verify a proof of valid inference for MobileNet v2 achieving 79% accuracy while simultaneously being verifiable in 10 seconds on commodity hardware. Furthermore, our

proving times can improve up to one to four orders of magnitude compared to prior work Feng et al. (2021); Weng et al. (2022); Lee et al. (2020); Liu et al. (2021). We further provide practical protocols leveraging these ZK-SNARKs to verify ML model accuracy, verify MP predictions, and use ML models for audits. These results demonstrate the feasibility of practical, verified ML model execution.

To ZK-SNARK ImageNet-scale models, we leverage recent developments in ZK-SNARK proving systems zcash (2022). Our key insight is that *off-the-shelf* proving systems for generic computation are sufficient for verified ML model execution, with careful translation from DNN specifications to ZK-SNARK arithmetic circuits. Furthermore, off-the-shelf proving systems allow for hardware acceleration, leading to substantially faster proving times. Our arithmetization uses two novel optimizations: lookup arguments for non-linearities and reuse of sub-circuits across layers (Section 4). Without our optimizations, the ZK-SNARK construction will require an impractically large amount of hardware resources.

ZK-SNARKs have several surprising properties (Section 3). Importantly, ZK-SNARKs allow portions of the input and intermediates to be kept hidden (while selectively revealing certain inputs) and are *non-interactive*. The non-interactivity allows third parties to trustlessly adjudicate disputes between MPs and MCs and verify the computation without participating in the computation itself.

In the setting of verified DNN inference, ZK-SNARKs allow us to hide the weights, the inputs, or both. The hidden portions can then be committed to by computing and revealing hashes of the inputs, weights, or both (respectively). In particular, an MP may be interested in keeping its proprietary weights hidden while being able to convince an MC of valid inference. The ZK-SNARK primitive allows the MP to commit to the (hidden) weights while proving execution.

Given the ability to ZK-SNARK ML models while committing to and selectively revealing chosen portions of their inputs, we propose methods of verifying MLaaS model accuracy, MLaaS model predictions, and trustless retrieval of documents in the face of *malicious* adversaries. Our protocols combine ZK-SNARK proofs and economic incentives to create trustless systems for these tasks. We further provide cost estimates for executing these protocols.

## 2 RELATED WORK

**Secure ML.** Recent work has proposed secure ML as a paradigm for executing ML models Ghodsi et al. (2017b); Mohassel & Zhang (2017); Knott et al. (2021). There are a wide range of security models, including verifying execution of a known model on untrusted clouds Ghodsi et al. (2017b), input privacy-preserving inference Knott et al. (2021), and weight privacy-preserving inference. The most common methods of doing secure ML are with multi-party computation (MPC), homomorphic encryption (HE), or interactive proofs (IPs). These methods are either impractical, do not work in the face of malicious adversaries Knott et al. (2021); Kumar et al. (2020); Lam et al. (2022); Mishra et al. (2020), or do not hide the weights/inputs Ghodsi et al. (2017b). In this work, we propose practical methods of verified ML execution in the face of malicious adversaries.

**MPC.** One method for secure ML is MPC, in which the computation is shared across multiple parties Knott et al. (2021); Kumar et al. (2020); Lam et al. (2022); Mishra et al. (2020); Jha et al. (2021); Weng et al. (2021). All MPC protocols have shared properties: they require interaction (i.e., both parties must be simultaneously online) but can perform computation without revealing the computation inputs (i.e., weights and ML model inputs) across parties.

There are several security assumptions for MPC protocols. The most common assumption is the *semi-honest adversary*, in which the malicious party participates in the protocol honestly but attempts to steal information. In this work, we focus on *malicious adversaries*, who can choose to deviate from the protocol. Unfortunately, MPC that is secure against malicious adversaries is impractical: it can cost up to 550 GB of communication and 657 seconds of compute per example on toy datasets Pentyala et al. (2021). In this work, we provide a practical, alternative method of verifying ML model inference in the face of malicious adversaries. Furthermore, our methods do not require per-example communication.

**HE.** Homomorphic encryption allows parties to perform computations on encrypted data without decrypting the data Armknecht et al. (2015). HE is deployed to preserve privacy of the inputs, but cannot verify that ML model execution happened correctly. Furthermore, HE is currently impractical

for ML models. Since ML model inference can take up to gigaflops of computation, HE has only been deployed on toy datasets such as MNIST or CIFAR-10 Lou & Jiang (2021); Juvekar et al. (2018).

**ZK-SNARKs for secure ML.** Some recent work has produced ZK-SNARK protocols for neural network inference on smaller datasets like MNIST and CIFAR-10. Some of these works like Feng et al. (2021) use older proving systems like Groth (2016). Other works Ghodsi et al. (2017a); Lee et al. (2020); Liu et al. (2021); Weng et al. (2022) use interactive proof or ZK-SNARK protocols based on sum-check Thaler (2013) custom-tailored to DNN operations such as convolutions or matrix multiplications. Compared to these works, our work in the modern Halo2 proving system zcash (2022) allows us to use the Plonkish arithmetization to more efficiently represent DNN inference by leveraging lookup arguments and well-defined custom gates. Combined with the efficient software package `halo2` and advances in automatic translation, we are able to outperform these methods.

## 3 ZK-SNARKS

**Overview.** Consider the task of verifying a function evaluation $y = f(x; w)$ with *public inputs* $x$, *private inputs* $w$, and output $y$. In the setting of public input and hidden model, $x$ is an image, $w$ is the weights of a DNN, and $y$ is the result of executing the DNN with weights $w$ on $x$.

A ZK-SNARK Bitansky et al. (2017) is a cryptographic protocol allowing a Prover to generate a proof $\pi$ so that with knowledge of $\pi$, $y$, and $x$ alone, a Verifier can check that the Prover knows some $w$ so that $y = f(x; w)$. ZK-SNARK protocols satisfy non-intuitive properties summarized informally below:

1. *Succinctness*: The proof size is sub-linear (typically constant or logarithmic) in the size of the computation (i.e., the complexity of $f$).

2. *Non-interactivity*: Proof generation does not require interaction between the verifier and prover.

3. *Knowledge soundness*: A computationally bounded prover cannot generate proofs for incorrect executions.

4. *Completeness*: Proofs of correct execution verify.

5. *Zero-knowledge*: $\pi$ reveals no information about $w$ beyond what is contained in the output and $x$.

Most ZK-SNARK protocols proceed in two steps. In the first step, called *arithmetization*, they produce a system of polynomial equations over a large prime field (an arithmetic circuit) so that finding a solution is equivalent to computing $f(x; w)$. Namely, for $(f, y, x, w)$, the circuit constraints are met if and only if $y = f(x; w)$. In the second step, a cryptographic *proof system*, often called a *backend*, generates a ZK-SNARK proof.

This work uses the Halo2 ZK-SNARK protocol zcash (2022) implemented in the `halo2` software package. In contrast to ZK-SNARK schemes custom designed for neural networks in prior work Liu et al. (2021); Lee et al. (2020), Halo2 is designed for general-purpose computation, and `halo2` has a broader developer ecosystem. This means we inherit the security, efficiency, and usability of the resulting developer tooling. In the remainder of this section, we describe the arithmetization and other properties of Halo2.

**Plonkish arithmetization.** Halo2 uses the Plonkish arithmetization zcash (2022), which allows polynomial constraints and certain restricted forms of randomness. It is a special case of a randomized AIR with preprocessing Ben-Sasson et al. (2018); Gabizon (2021) which unifies recent proof systems Gabizon et al. (2019); Gabizon & Williamson (2020); Pearson et al. (2022).

Variables in the arithmetic circuit are arranged in a rectangular grid with cells valued in a 254-bit prime field. The Plonkish arithmetization allows three types of constraints:

*Custom gates* are polynomial expressions over cells in a single row that must vanish on *all* rows of the grid. As an example, consider a grid with columns labeled $a, b, c$ with $a_i, b_i, c_i$ being the cells in row $i$. The custom *multiplication gate* $a_i \cdot b_i - c_i = 0$ enforces that $c_i = a_i \cdot b_i$ for all rows $i$.

In nearly all circuits, it is beneficial to have custom gates only apply to specific rows. To do this, we can add an extra column $q$ (per custom gate), where each cell in $q$ takes the value 0 or 1; $q$ is called a selector. Then, we can modify the custom gate to only apply for rows where $q_i \neq 0$: $q_i \cdot (a_i \cdot b_i - c_i) = 0$.

*Permutation arguments* allow us to constrain pairs of cells to have equal values. They are used to copy values from one cell to another. They are implemented via randomized polynomial constraints for *multiset* equality checks.

*Lookup arguments* allow us to constrain a $k$-tuple of cells $(d_i^1, \ldots, d_i^k)$ in the same row $i$ with the following constraint. For a disjoint set of $k$ other columns, the cells $(d_i^1, \ldots, d_i^k)$ must match the values in some other row $i'$. Namely, we can constrain $(d_i^1, \ldots, d_i^k) = (e_{i'}^{1'}, \ldots, e_{i'}^{k'})$. This constrains $(d_i^1, \ldots, d_i^k)$ to lie in the *lookup table* defined by those $k$ other columns. We use lookup arguments in the arithmetization in two ways. First, we implement range checks on a cell $c$ by constraining it to take values in a fixed range $\{0, \ldots, N-1\}$. Second, we implement non-linearities by looking up a pair of cells $(a, b)$ in a table defined by exhaustive evaluation of the non-linearity. Lookup arguments are also implemented by randomized polynomial constraints.

Prior work on SNARK-ing neural networks using proof systems intended for generic computations started with the more limited R1CS arithmetization Gennaro et al. (2013) and the Groth16 proof system Groth (2016), in which DNN inference is less efficient to express. In Section 4, we describe how to use this more expressive Plonkish arithmetization to efficiently express DNN inference.

**Performance for Halo2 circuits.** Halo2 is a polynomial interactive oracle proof (IOP) Ben-Sasson et al. (2016) made non-interactive via the Fiat-Shamir heuristic. In a polynomial IOP, the ZK-SNARK is constructed from column polynomials which interpolate the values in each column. In Halo2, these polynomials are fed into the *inner product argument* introduced by Bowe et al. (2019) to generate the final ZK-SNARK.

Several aspects of performance matter when evaluating a ZK-SNARK proof for a computation. First, we wish to minimize proving time for the Prover and verification time for the Verifier. Second, on both sides, we wish to minimize the proof size. All of these measures increases with the number of rows, columns, custom gates, permutation arguments, and lookup arguments.

# 4 CONSTRUCTING ZK-SNARKS FOR IMAGENET-SCALE MODELS

We now describe our main contribution, the implementation of a ZK-SNARK proof for MobileNetv2 inference Sandler et al. (2018) in `halo2`. This requires arithmetizing the building block operations in standard convolutional neural networks (CNNs) in the Plonkish arithmetization.

## 4.1 ARITHMETIZATION

Standard CNNs are composed of six distinct operations: convolutions, batch normalization, ReLUs, residual connections, fully connected layers, and softmax. We fuse the batch normalization into the convolutions and return the logits to avoid executing softmax. We now describe our ingredients for constraining the remaining four operations.

**Quantization and fixed-point.** DNN inference is typically done in floating-point arithmetic, which is extremely expensive to emulate in the prime field of arithmetic circuits. To avoid this overhead, we focus on DNNs quantized in `int8` and `uint8`. For these DNNs, weights and activations are represented as 8-bit integers, though intermediate computations may involve up to 32-bit integers.

In these quantized DNN, each weight, activation, and output is stored as a tuple $(w_{\text{quant}}, z, s)$, where $w_{\text{quant}}$ and $z$ is an 8-bit integer weight and zero point, and $s$ is a floating point scale factor. $z$ and $s$ are often shared for all weights in a layer, which reduces the number of bits necessary to represent the DNN. In this representation, the weight $w_{\text{quant}}$ represents the real number weight: $w = (w_{\text{quant}} - z) \cdot s$.

To efficiently arithmetize the network, we replace the floating point $s$ by a fixed point approximation $\frac{a}{b}$ for $a, b \in \mathbb{N}$ and compute $w$ via $w = ((w_{\text{quant}} - z) \cdot a)/b$. The intermediate arithmetic is done in standard 32-bit integer arithmetic. Our choice of lower precision values of $a$ and $b$ results in a slight accuracy drop but dramatic improvements in prover and verifier performance.

As an example of fixed point arithmetic after this conversion, consider adding $y = x_1 + x_2$ with zero points and scale factors $z_y, z_1, z_2$ and $s_y, s_1, s_2$, respectively. The floating point computation

$$(y - z_y) \cdot s_y = (x_1 - z_1) \cdot s_1 + (x_2 - z_2) \cdot s_2$$

is replaced by the fixed point computation

$$y \approx (x_1 - z_2) \cdot \frac{a_1}{b_1}\frac{b_y}{a_y} + (x_2 - z_2) \cdot \frac{a_2}{b_2}\frac{b_y}{a_y} + z_y.$$

The addition and multiplication can be done natively in the field, but the division cannot. To address this, we factor the computation of each layer into dot products and create a custom gate to verify division. We further fuse the division and non-linearity gates for efficiency.

**Custom gates for linear layers.** MobileNets contain three linear layers (layers with only linear operations): convolutions, residual connections, and fully connected layers. For these linear layers, we perform the computation per activation. To avoid expensive floating point scaling by the scale factor and the non-linearities, we combine these operations into a single sub-circuit.

To reduce the number of custom gates, we only use two custom gates for all convolutions, residual connections, and fully connected layers. The first custom gate constrains the addition of a fixed number of inputs $x_i^j$ in row $i$ via $c_i = \sum_{j=1}^{N} x_i^j$. The second custom gate constrains a dot product *of fixed size* with a zero point. For constant zero point $z$, inputs $x_i^j$, weights $w_i^j$, and output $c_i$ in row $i$, the gate implements the polynomial constraint

$$c_i = \sum_{j=1}^{N}(x_i^j - z) \cdot w_i^j$$

for a fixed $N$. To implement dot products of length $k < N$, we constrain $w_{k+1},...,w_N = 0$. For dot products of length $k > N$, we use copy constraints and the addition gate.

**Lookup arguments for non-linearities.** Consider the result of an unscaled, flattened convolution: $c_i = \sum_j x_i^j \cdot w_i^j$, where $j$ indexes over the image height, width, and channels, and $i$ is the row. Performing scale factor division and (clipped) ReLU to obtain the final activation requires computing

$$a_i = \text{ClipAndScale}(c_i, a; b) := \text{clip}\left(\left\lfloor \frac{c_i \cdot a}{b} \right\rfloor, 0, 255\right).$$

To constrain this efficiently, we apply a lookup argument and use the same value of $b$ across layers. To do so, we first perform the division by $b$ using a custom gate. Since $b$ is fixed, we can use the same custom gate and lookup argument. Let $d_i = \frac{c_i \cdot a}{b}$. We then precompute the possible values of the input/output pairs of $(d_i, a_i)$ to form a lookup table $T = \{(c, \text{ClipAndScale}(c)) \mid c \in \{0, ..., N\}\}$. $N$ is chosen to cover the domain, namely the possible values of $c$. We then use a lookup argument to enforce the constraint $\text{Lookup}[(d_i, a_i) \in T]$.

We emphasize that naively using lookup arguments would result in a different lookup argument per layer since the scale factors differ. Using different lookup arguments would add high overhead, which our approach avoids. Sharing lookups can result in small accuracy losses (e.g., as little as 0.1%) but can save over $2\times$ the computational burden.

## 4.2 COMMITTING TO WEIGHTS OR INPUTS

ZK-SNARKs allow parts of the inputs to be made public, in addition to revealing the outputs of the computation (Section 3). For ML models, the input (e.g., image), weights, or both can be made public. Then, to commit to the hidden inputs, the hash can be computed within the ZK-SNARK and be made public. Concretely, we use the following primitives: 1) hidden input, public weights, 2) public input, hidden weights, and 3) hidden input, hidden weights. Commitments to all hidden parts are made public. We use an existing circuit for the SNARK-friendly Poseidon hash Grassi et al. (2019).

## 5 APPLICATIONS OF VERIFIED ML MODEL INFERENCE

Building upon our efficient ZK-SNARK constructions, we now provide protocols to verify ML model accuracy, verify ML model predictions for serving, and trustlessly retrieve documents matching a predicate based on an ML model.

## 5.1 PROTOCOL PROPERTIES AND SECURITY MODEL

**Protocol properties.** In this section, we describe and study the properties of protocols leveraging verified ML inference. Each protocol has a different set of requirements, which we denote $A$. The requirements $A$ may be probabilistic (e.g., the model has accuracy 80% with 95% probability). We are interested in the *validity* and *viability* of our protocols. Validity that if the protocol completes, $A$ holds. Viability refers to the property that rational agents will participate in the protocol.

**Security model.** In this work, we use the standard ZK-SNARK security model for the ZK-SNARKs Bünz et al. (2020). Informally, the standard security model states the prover and verifier only interact via the ZK-SNARKs and that the adversary is computationally bounded, which excludes the possibility of side channels. Our security model allows for malicious adversaries, which is in contrast to the semi-honest adversary setting. Recall that in the semi-honest adversary setting, the adversaries honestly follow the protocol but attempt to compromise privacy.

**Assumptions.** For validity, we only assume two standard assumptions. First, that it is hard to compute the order of random group elements Bünz et al. (2020), as implied by the RSA assumption Rivest et al. (1978). Second, that finding hash collisions is difficult Rogaway & Shrimpton (2004). Only requiring cryptographic hardness assumptions is sometimes referred to as *unconditional* Ghodsi et al. (2017a).

For viability, we assume the existence of a programmatic escrow service and that all parties are economically rational. In the remainder of this section, we further assume the "no-griefing condition," which states that no party will purposefully loses money to hurt another party, and the "no-timeout condition," which states that no parties will time out. Both of these conditions can be relaxed. We describe how to relax these conditions in the Appendix.

## 5.2 VERIFYING ML MODEL ACCURACY

In this setting, a model consumer (MC) is interested in verifying a model provider's (MP) model's accuracy, and MP desires to keep the weights hidden. As an example use case, MC may be interested in verifying the model accuracy to purchase the model or to use MP as an ML-as-a-service provider (i.e., to purchase predictions in the future). Since the weights are proprietary, MP desires to keep the weights hidden. The MC is interested in *verifiable* accuracy guarantees, to ensure that the MP is not lazy, malicious, or serving incorrect predictions. In this section, we use concrete constants for ease of analysis but show that they can be varied in the Appendix.

Denote the cost of obtaining a test input and label to be $E$, the cost of ZK-SNARKing a single input to be $Z$, and $P$ to be the cost of performing inference on a single data point. We enforce that $E > Z > P$. Furthermore, let $N = N_1 + N_2$ be the number of examples used in the verification protocol, where $N_1$ is set based on the economic value of the model and $N_2$ is set such that $N$ is large enough to identify the model accuracy within a desired precision. These parameters are marketplace-wide and are related to the security of the protocol.

The protocol requires that MP stakes $1000N_1E$ per model to participate. The stake is used to prevent Sybil attacks, in which a single party fakes the identity of many MPs. Given the stake, the verification protocol is as follows for some accuracy target $a$:

1. MP commits to an architecture and weights (by providing the weight hash). MC commits to a test set $\{(x_1,y_1),...,(x_N,y_N)\}$ by publishing the hash of the examples.

2. MP and MC escrows $2NE+\epsilon$, where $\epsilon$ goes to the escrow service.

3. MC sends the test set to MP. MP can continue or abort. If MP aborts, MC loses $NP$ of the escrow.

4. MP sends ZK-SNARKs and the outputs of the model on the test set to MC.

5. If accuracy target $a$ is met, MC pays $2NZ$. Otherwise, MP loses the full amount $2NE$ to MC.

The verification protocol is valid because MP must produce the outputs of the ML model as enforced by the ZK-SNARKs. MC can compute the accuracy given the outputs. Thus, if the protocol completes, the accuracy target is met.

If the economic value of the transaction exceeds $1000N_1E$, the protocol is viable since the MP will economically benefit by serving or selling the model. This follows as we have chosen the stake

parameters so that malicious aborting will cost the MC or MP more in expectation than completing the protocol. We formalize our analysis and give a more detailed analysis in the Appendix.

## 5.3 VERIFYING ML MODEL PREDICTIONS

In this setting, we assume that MC has verified model accuracy and is purchasing predictions in the ML-as-a-service setting. As we show, MC need not request a ZK-SNARK for every prediction to bound malicious MP behavior.

The serving verification procedure proceeds in rounds of size $K$ (i.e., prediction is served over $K$ inputs). MC is allow to contest at any point during the round, but not after the round has concluded. Furthermore, let $K \geq K_1 > 0$. The verification procedure is as follows:

1. MC escrows $2KZ$ and MP escrows $\beta KZ$, where $\beta \geq 2$ is decided between MP and MC.

2. MC provides hashes for the $K$ inputs to the escrow and sends $x_i$ to MP. MP verifies the hashes.

3. MP provides the predictions $(y_i)$ to the inputs (without ZK-SNARKs) to MC. MC provides the hash of $\text{Concat}(x_i, y_i)$ to the escrow.

4. If MC believes MP is dishonest, MC can contest on any subset $K_1$ of the predictions.

5. When contested, MP will provide the ZK-SNARKs for the $K_1$ predictions. If MP fails to provide the ZK-SNARKs, then it loses the full $\beta ZP$.

6. If the ZK-SNARKs match the hashes, MC loses $2K_1Z$ from the escrow and the remainder of the funds are returned. Otherwise, MP loses the full $\beta ZP$ to MC.

For validity, if MP is honest, MC cannot contest successfully and the input and weight hashes are provided. Similarly, if MC is honest and contests an invalid prediction, MP will be unable to produce the ZK-SNARK.

For viability, first consider an honest MP. The honest MP is indifferent to the escrow as it receives the funds back at the end of the round. Furthermore, all contests by MC will be unsuccessful and MP gains $K_1Z$ per unsuccessful contest.

For honest MC to participate, they must either have a method of detecting invalid predictions with probability $p$ or they can randomly contest a $p$ fraction of the predictions. Note that for random contests, $p$ depends on the negative utility of MC receiving an invalid prediction. As long as $\beta KZ$ is large relative to $\frac{KZ}{p}$, then MC will participate.

## 5.4 TRUSTLESS RETRIEVAL OF ITEMS MATCHING A PREDICATE

In this setting, a requester wishes to retrieve records that match the output of an ML model (i.e., a predicate) from a responder. These situations often occur during legal subpoenas, in which a judge requires the responder to send a set of documents matching the predicate. For example, the requester may be a journalist requesting documents under the Freedom of Information Act or the plaintiff requesting documents for legal discovery. This protocol could also be useful in other settings where the responder wishes to prove that a dataset does not contain copyrighted content.

When a judge approves this request, the responder must divulge documents or images matching the request. We show that ZK-SNARKs allow requests encoded as ML algorithms to be trustlessly verified. The protocol proceeds as follows:

1. The responder commits to the dataset by producing hashes of the documents.

2. The requester sends the model to the responder.

3. The responder produces ZK-SNARKs of the model on the documents, with the inputs hashed. The responder sends the requester the documents that match the positive class of the model.

The audit protocol guarantees that the responder will return the documents from Stage 1 that match the model's positive class. The validity follows from the difficulty of finding hash collisions and the security of ZK-SNARKs. The responder may hash invalid documents (e.g., random or unrelated images), which the protocol makes no guarantees over. This can be mitigated based on whether the documents come from a trusted or untrusted source.

| Model | Accuracy (top-5) | Setup time | Proving time | Verification time | Proof size (bytes) |
|---|---|---|---|---|---|
| MobileNet, 0.35, 96 | 59.1% | 49.1 s | 92.0 s | 0.23 s | 5984 |
| MobileNet, 0.5, 224 | 75.7% | 455.5 s | 831.9 s | 2.32 s | 7008 |
| MobileNet, 0.75, 192 | 79.2% | 571.7 s | 850.7 s | 3.23 s | 5408 |

Table 1: Accuracy, setup time, proving time, and verification time of MobileNet v2s. The first parameter is the "expansion size" parameter for the MobileNet and the second parameter is image resolution. It is now possible to SNARK ImageNet models, which no prior work can achieve.

| Method | Proving time lower bounds (s) |
|---|---|
| Zen | 20,000 |
| vCNN | 172,800 |
| pvCNN | 31,011[*] |
| zkCNN | 1,597[*] |

Table 2: Lower bounds on the proving time for prior work. For Zen and vCNN, we compared against a DNN with strictly fewer operations compared to MobileNet v2 (0.35, 96). For pvCNN and zkCNN, we estimate the lower bound by scaling the computation.

Documents from a trusted source can be verified from the trusted source digitally signing the hashes. As an example, hashes for government-produced documents (in the FOIA setting) may be produced at the time of document creation.

For documents from an untrusted source, we require a commitment for the entire corpus. Given the commitment, the judge can allow the requester to randomly sample a small number ($N$) of the documents to verify the hashes. The requester can verify that the responder tampered with at most $p = \exp\left(\frac{1-\delta}{N}\right)$ for some confidence level $\delta$.

# 6 EVALUATION

To evaluate our ZK-SNARK system, we ZK-SNARKed MobileNets with varying configurations. We evaluated the hidden model and hidden input setting, the most difficult setting for ZK-SNARKs.

We measured: model accuracy, setup time, proving time, and verification time. The setup time is done once per MobileNet and is independent of the weights. The proving is done by the model provider and the verification is done by the model consumer. Proving and verification must be done once per input. To the best of our knowledge, no prior work can ZK-SNARK DNNs on ImageNet scale models.

We ZK-SNARK quantized DNNs (which avoids floating-point computations) as provided by TensorFlow Slim Silberman & Guadarrama (2018). MobileNet v2 has two adjustable parameters: the "expansion size" and the input dimension. We vary these parameters to see the effect on the ZK-SNARKing time and accuracy of the models.

## 6.1 ZK-SNARKS FOR IMAGENET-SCALE MODELS

We first present results when creating ZK-SNARKs for only the DNN execution, which all prior work on ZK-SNARKs for DNNs do. Namely, we do not commit to the model weights in this section.

We summarize results for various MobileNet v2 configurations in Table 1. As shown, we can achieve up to 79% accuracy on ImageNet, while simultaneously taking as few as 3.23s and 5408 bytes to verify. Furthermore, the ZK-SNARKs can be scaled down to take as few as 0.7s to verify at 59% accuracy. These results show the feasibility of ZK-SNARKing ImageNet-scale models.

In contrast, we show lower bounds on the proving time for prior work on a comparable model (MobileNet v2 (0.35, 96)). We were unable to reproduce any of the prior work, but we use the proving numbers presented in the papers. For Zen, and vCNN we use the largest model in the respective papers as lower bounds (MNIST or CIFAR10 models). For zkCNN and pvCNN we estimate the proving time by scaling the largest model in the paper. As shown in Table 2, the proving time for the prior work is at least $17\times$ higher than our method and up to $1,800\times$ higher. We emphasize that these are lower bounds on the proving time for prior work.

| Fraction | Sample size | Cost |
|---|---|---|
| 5% | 72 | $7.65 |
| 2.5% | 183 | $19.43 |
| 1% | 366 | $38.86 |

Table 3: Costs of performing verified prediction and trustless retrieval while bounding the fraction of predictions tampered with. Cost were estimated with the MobileNet v2 (0.35, 96) model.

| $\epsilon$ | Sample size | Total cost |
|---|---|---|
| 5% | 600 | $63.71 |
| 2.5% | 2,396 | $254.4 |
| 1% | 14,979 | $1590.5 |

Table 4: Cost of verifying the accuracy of an ML model within some $\epsilon$ of the desired accuracy. Costs were estimated with the MobileNet v2 (0.35, 96) model.

Finally, we note that the proof sizes of our ZK-SNARKs are orders of magnitude less than MPC methods, which can take tens to hundreds of gigabytes.

## 6.2 PROTOCOL EVALUATION

We present results when instantiating the protocols described in Section 5. To do so, we ZK-SNARK MobileNet v2 (0.35, 96) *while* committing to the weights, which *no other prior work does*. For the DNNs we consider, the cost of committing to the weights via hashes is approximately the cost of inference itself. This phenomenon of hashing being proportional to the computation cost also holds for other ZK-SNARK applications Privacy & Explorations (2022).

For each protocol, we compute the cost using public cloud hardware for the prover and verifier for a variety of protocol parameters.

**Verifying prediction, trustless retrieval.** For both the verifying predictions and trustless retrieval, the MC (requester) can bound the probability that the MP (responder) returns incorrect results by random sampling. If a single incorrect example is found, the MC (requester) has recourse. In the verified predictions setting, MC will financially gain and in the retrieval setting, the requester can force the judge to make the responder turn over all documents.

The MC can choose a confidence level $\delta$ and a bound on the fraction of predictions tampered $p$. The MC can then choose a random sample of size $N$ as determined by inverting a valid Binomial proportion confidence interval. Namely, $N$ *is independent of the size of the batch*.

We compute the number of samples required and the cost of the ZK-SNARKs (both the proving and verifying) at various $p$ at $\delta = 5\%$, with results in Table 3. We use the Clopper-Pearson exact interval Clopper & Pearson (1934) to compute the sample size.

To contextualize these results, consider the Google Cloud Vision API which charges $1.50 per 1,000 images. Predictions over one million images would cost $1,500. If we could scale ZK-SNARKs to verify the Google API model with cost on par with MobileNet v2 (0.35, 96), verifying these predictions would add 2.4% overhead, which is acceptable in many circumstances.

**Verifying model accuracy.** For verifying MP model accuracy, the MC is interested in bounding probability that the accuracy target $a$ is not met $P(a' < a) \leq \delta$ for the estimated accuracy $a'$ and some confidence level $\delta$. We focus on binary accuracy in this evaluation. For binary accuracy, we can use Hoeffding's inequality to solve for the sample size: $P(a - a' > \epsilon) \leq \exp(-2\epsilon^2/N) = \delta$.

We show the total number of samples needed for various $\epsilon$ at $\delta = 5\%$ and the costs in Table 4. Although these costs are high, they are within the realm of possibility. For example, it may be critical to verify the accuracy of a financial model or a model used in healthcare settings. For reference, even moderate-size datasets can cost on the order of $85,000 Incze (2019), so verifying the model would add between 0.07% to 1.9% overhead compared to just the cost of obtaining training data.

## 7 CONCLUSION

In this work, we present protocols for verifying ML model execution trustlessly for audits, testing ML model accuracy, and ML-as-a-service inference. We further present the first ZK-SNARKed ImageNet-scale model to demonstrate the feasibility of our protocols. Combined, our results show the promise for verified ML model execution in the face of malicious adversaries.

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

## A  VIABILITY OF VERIFYING MODEL ACCURACY

In this section, we prove the viability of the simplified protocol for verifying model accuracy.

As mentioned, viability further requires that the cost of the model or price of post-verification purchased predictions is greater than $1000N_1E$. Viability requires that honest MP/MC will participate and that dishonest MP/MC will not participate.

Consider the case of an honest MP. If MC is dishonest, it can economically gain by having MP proceed beyond Stage 4 and having MP fail the accuracy target. However, as MP has access to the test set, they can determine the accuracy before proceeding beyond 4, so will not proceed if the accuracy target is not met. If MP has a valid model, they will proceed, since the profits of serving predictions or selling the model is larger than their stake.

Consider the case of an honest MC. Note that an economically rational MP is incentivized to serve the model if it has a model of high quality. Thus, we assume dishonest MPs do not have model that achieves the accuracy target. The dishonest MP can economically gain by aborting at Stage 4 at least 1000 times (as $E > P$). MC can choose to participate with MP that only has a failure rate of at most 1%. In order to fool honest MCs, MP must collude to verify invalid test sets, which costs $2\epsilon$ per verification. MP must have 99 fake verifications for one failed verification from an honest MC. Thus, by setting $\epsilon = \frac{NP}{99}$, dishonest MP will not participate.

From our analysis, we see that honest MP and MC are incentivized to participate and that dishonest MP and MC will not participate, showing viability.

## B  VERIFYING ML MODEL ACCURACY WITH GRIEFING AND TIMEOUTS

In this section, we describe how to extend our model accuracy protocol to account for griefing and timeouts. Griefing is when an adversarial party purposefully performs economically disadvantageous actions to harm another party. Timeouts are when either the MP or MC does not continue with the protocol (whether by choice or not) without explicitly aborting.

Denote the cost of obtaining a test input and label to be $E$, the cost of ZK-SNARKing a single input to be $Z$, and $P$ to be the cost of performing inference on a single data point. We enforce that $E > Z > P$. Furthermore, let $N = N_1 + N_2$ be the number of examples used in the verification protocol. These parameters are marketplace-wide and are related to the security of the protocol.

The marketplace requires MP to stake $1000N_1E$ per model to participate. The stake is used to prevent Sybil attacks, in which a single party fakes the identity of many MPs. Given the stake, the verification protocol is as follows for some accuracy target $a$:

1. MP commits to an architecture and set of weights (by providing the ZK-SNARK keys and weight hash respectively). MC commits to a test set $\{(x_1,y_1),...,(x_N,y_N)\}$ by publishing the hash of the examples.

2. MP and MC escrows $2NE+\epsilon$, where $\epsilon$ goes to the escrow service.

3. MP selects a random subset of size $N_1$ of the test set. If MC aborts at this point, MC loses the full amount in the escrow to MP. If MC continues, it sends the subset of examples to MP.

4. MP chooses to proceed or abort. If MP aborts, MC loses $N_1P$ of the escrow to MP and the remainder of the funds are returned to MC and MP.

5. MC sends the remainder of the $N_2$ examples to MP. If MP aborts from here on out, MP loses the full amount in the escrow ($2NE$) to MC.

6. MP sends SNARKs of the $N_2$ examples with outputs revealed. The weights and inputs are hashed.

7. If accuracy target $a$ is met, MC pays $2(N_1P+N_2Z)$. Otherwise, MP loses the full amount $2NE$ to MC.

**Validity and viability (no griefing or timeouts).** The verification protocol is valid because MP must produce the outputs of the ML model as enforced by the ZK-SNARKs. MC can compute the accuracy given the outputs. Thus, if the protocol completes, the accuracy target is met.

Viability further requires that the cost of the model or price of post-verification purchased predictions is greater than $1000N_1E$. We must show that honest MP/MC will participate and that dishonest MP/MC will not participate. We first show viability without griefing or timeouts and extend our analysis below.

Consider the case of an honest MP. If MC is dishonest, it can economically gain by having MP proceed beyond Stage 4 and having MP fail the accuracy target. Since MP chooses the subsets $N_1$ and $N_2$, they can be drawn uniformly from the full test set. Thus, MP can choose to proceed only if $P(a \text{ met}|N_1) > 1 - \alpha$ is such that expected value for MP is positive, where $\alpha$ depends on the choice of $\epsilon$ (we provide concrete instantiations for $\alpha$ and $\epsilon$ below). If MC is honest, MP gains in expected value by completing the protocol, as its expected gain is

$$(1-\alpha)(N_1P + 2N_2Z - \epsilon) + \alpha N_1 P.$$

Consider the case of an honest MC. Note that an economically rational MP is incentivized to serve the model if it has a model of high quality. Thus, we assume dishonest MPs do not have model that achieves the accuracy target. The dishonest MP can economically gain by aborting at Stage 4 at least 1000 times (as $E > P$). MC can choose to participate with MP that only has a failure rate of at most 1%. In order to fool honest MCs, MP must collude to verify invalid test sets, which costs $2\epsilon$ per verification. MP must have 99 fake verifications for one failed verification from an honest MC. Thus, by setting $\epsilon = \frac{N_1P}{99}$, dishonest MP will not participate. For this choice of $\epsilon$, $\alpha > \frac{49N_1P}{49N_1P+99NE}$.

From our analysis, we see that honest MP and MC are incentivized to participate and that dishonest MP and MC will not participate, showing viability.

**Accounting for griefing.** We have shown that there exist choices of $\alpha$ and $\epsilon$ for viability with economically rational actors. However, we must also account for griefing, where an economically irrational actor harms themselves to harm another party. It is not possible to making griefing impossible. However, we can study the costs of griefing. By making these costs high, our protocol will discourage griefing. In order to make these costs high, we let $\epsilon = N_1P$.

We first consider griefing attacks against MC. For the choice of $\epsilon$, dishonest MP must pay $99N_1P$ per honest MC it griefs. In particular, MC loses $N_1P$ per attack, so the cost of a griefing MP is $99\times$ higher than the cost to MC.

We now consider griefing attacks against an MP. Since MP can randomly sample, MP can simply choose $\alpha$ appropriately to ensure the costs to a griefing MC is high. In particular, the MP pays $2NE$ per successful attack. MP's expected gain for executing the protocol is

$$(1-\alpha)(2N_2Z) + \alpha N_1 P$$

for the choice of $\epsilon$ above. Then, for

$$\alpha = \frac{\frac{1}{50}NE - 2N_2Z}{N_1P - N_2Z}$$

the cost of griefing is $100\times$ higher for griefing MC than MP. By choosing $N_1$ and $N_2$ appropriately, MP can ensure the cost of griefing is high for griefing MCs.

**Accounting for timeouts.** Another factor to consider is that either MC or MP can choose not to continue the protocol without explicitly aborting. To account for this, we introduce a sub-protocol for sending the data. Once the data is sent, if MP does not continue after time period of time, MP is slashed.

The sub-protocol for data transfer is as follows:

1. MC sends hashes of encrypted inputs to escrow and MP.

2. MC sends encrypted inputs to MP.

3. MP signs and publishes an acknowledgement of the receipt.

4. MC publishes decryption key.

5. MP contests that the decryption key is invalid or continues the protocol.

If MC does not respond or aborts in Stages 1, 2, or 4, it is slashed. If MP does not respond in Stages 3 or 5, it is slashed.

Validity follows from standard cryptographic hardness assumptions. Without the decryption key, MP cannot access the data. With the decryption key, MP can verify that the data was sent properly.

## C    EVALUATION HARDWARE

For the evaluation, we used the smallest Amazon Web Services (AWS) instance type of the family `g4dn` that could prove each MobileNet v2 ZK-SNARK. For MobileNet v2 (0.35, 96), we used the `g4dn.8xlarge` instance type. For MobileNet v2 (0.5, 224), we used the `g4dn.16xlarge` instance type. For MobileNet v2 (0.75, 192), we used the `g4dn.metal` instance type.

## D    CODE

We have anonymized our code here: `https://anonymous.4open.science/r/zkml-72D8/README.md`

