# OpenReview forum: "Scaling up Trustless DNN Inference with Zero-Knowledge Proofs"
_ICLR.cc/2024/Conference — Submitted to ICLR 2024_

### Official Review · Reviewer_EWEY · 2023-10-26

**Soundness:** 3 good
**Presentation:** 3 good
**Contribution:** 2 fair
**Rating:** 6
**Confidence:** 3

**Summary:**

As machine learning demands a lot nowadays, users tend to outsource ML training and inference to clouds, which is MLaaS. However, users may think that most of cloud services are untrusted, dishonest or inaccurate, and they would like to verify the inference results from cloud servers. In this work, authors apply a zero-knowledge proofs technique ZK-SNARKs for verifiable ML inference in the untrusted setting. The ZK-based circuits created in this work can scale to large models compared to previous works.

**Strengths:**

1. Unlike most of secure machine learning, this work considers untrusted setting against malicious adversaries with better empirical performance on large-scale dataset, which makes this work broader scope in the real-world application.
2. Lookup arguments benefit and speed up evaluation of non-linearity.

**Weaknesses:**

1. Instead of an innovative work, this work looks more like an incremental work on secure ML with ZK-SNARKs, where applies some specific engineering tricks on MobileNet.
2. Although authors mention the Halo 2 is a general-purpose system, this work only focuses on the empirical results on image tasks.

**Questions:**

1. Do you consider interactive ZK proofs for the truthless DNN inference? or why do you think non-interactive is better? Online proofing could probably scale up more because interactive proofs could transmit circuits gradually instead of at once, especially for memory-bounded application.

---

### Official Review · Reviewer_hiWG · 2023-10-30

**Soundness:** 3 good
**Presentation:** 2 fair
**Contribution:** 2 fair
**Rating:** 5
**Confidence:** 2

**Summary:**

The paper discusses the challenges faced by consumers in verifying predictions from Machine Learning as a Service (MLaaS) platforms due to potentially malicious, lazy, or erroneous service providers. The paper claims the first practical method to non-interactively verify ML model inferences at ImageNet-scale, employing ZK-SNARKs (zero-knowledge succinct non-interactive argument of knowledge) for this purpose. The authors successfully demonstrate a ZK-SNARK proof for a full-resolution ImageNet model with 79% top-5 accuracy, where the verification process takes merely one second. The study further extends the use of ZK-SNARKs to design protocols for validating MLaaS predictions and accuracy.

**Strengths:**

The topic is new, interesting, and practical. The implementation seems clear and effective.

**Weaknesses:**

1. The writing is rather rough and there are many terms/statements that are unclear to me (please refer to Questions).
2. The contribution of this paper is unclear to me. It seems like the original ZK-SNARK (and its variants) is infeasible for deep NNs like MobileNet over large-scale datasets (Imagenet) but the authors didn’t explain the reason.  The authors should highlight the difficulties when verifying MobileNet/Imagenet with  ZK-SNARKs and how this paper solves these issues so that the contributions of this paper will be clearer.
3. The proposed adaptions to MobileNet only support 6 basic conv components, while more advanced techniques (e.g., attention mechanism) seem inapplicable. I believe this limits the deployments in the real world.

**Questions:**

1. For “Quantization and fixed-point”, what’s the performance drop after this quantization? Do you need model fine-tuning after quantization (cause a lot of quantization methods require fine-tuning)?
2. For “custom gate for linear layers”, the authors “use two custom gates for all convolutions, residual connections, and fully connected layers”. Does this bring additional model fine-tuning or performance drop?
3. What’s the meaning of “escrows”?
4. Could you please provide any attacker/malicious scenarios that show the effectiveness of your method? Such as backdoored MobileNet or MobileNet that generate random predictions with the probability of p.

---

### Official Review · Reviewer_U2XN · 2023-11-01

**Soundness:** 2 fair
**Presentation:** 2 fair
**Contribution:** 2 fair
**Rating:** 3
**Confidence:** 3

**Summary:**

In this paper, the author put forward an ImageNet-scale method for verifiable machine learning inference, where the integrity of the execution of machine learning is guaranteed. This method leverages zk-SNAKRs (Zero-Knowledge Succinct Non-Interactive Argument of Knowledge), cryptographic protocols that allow one party to prove the trueness of a given statement to another party without revealing any information about the statement itself except for its veracity. The author first translated the computation in deep neural networks, such as convolution, matrix multiplication, and ReLU into intermediate representations (also known as constraint systems). The zk-SANKRs proving system is run on the above intermediate representations to prove the integrity of machine learning inference. The author further put forward several application scenarios where such verified machine learning inference can be useful. The proposed method can verify the correctness of an inference of MobileNet-v2 on ImageNet within 10 seconds while achieving 79% top-5 accuracy.

**Strengths:**

The paper presents the critical problem of integrity of machine learning inference. It also implements a proving system with cryptographic tools, which can prove the correct execution of MobileNet-v2 on ImageNet.

**Weaknesses:**

(1) Novelty is limited. Firstly, the introduction of halo2 [a] and Plonkish [b] intermediate representation in section 3 is not the contribution of this paper. Secondly, the quantization method in section 4.1 is largely based on existing quantization methods[c], [f]. Thirdly, any method that achieves verifiable machine learning can be applied to the application scenarios in section 5.

(2) Critical inconsistency. There are multiple references to the performance of the proposed method that are inconsistent. For example, in the abstract, the authors mention “achieving 79% top-5 accuracy, with verification taking as little as one second”; in the introduction, it becomes “MobileNet v2 achieving 79% accuracy while simultaneously being verifiable in 10 seconds”. However, in the experiment part, in Table 1, the verification time for the model with 79% accuracy is reported as 3.23s. Such inconsistency weakens the contribution of this paper and makes it less reliable.

(3) Wrong or unsupported technical statements. The paper offers technical claims that are inaccurate, weakening its overall contribution. It is not true that [d], [e] are based on interactive protocols or sum-check protocols. They are based on Groth16 [h], which is a non-interactive protocol.

(4) Lack of evidence. (i)Top-1 accuracy should be reported. (ii)There should also be a comparison with respect to accuracy, verification time, and proving time with baseline methods. (iii)There is no evidence to support the claim “Our choice of lower precision values of a and b results in a slight accuracy drop but dramatic improvements in prover and verifier performance” in section 4.1.

(5) More comparison and contrast with the existing works are needed. For example, the method in [f] already scales to VGG16, which typically has 138M parameters, while the MobileNet V2 this paper focuses on typically has 3.5M parameters. More comparison with prior works is needed, with respect to the nature of the underlying cryptographic protocol such as interactivity, the proving time, and verification time.

(6) Justification for the choice of model is needed. There should be a reason why MobileNet V2 is studied instead of the commonly used VGG16 or ResNet18. Is MobileNet V2 more suitable for cloud computing? The top-5 accuracy from 59.1%~79.2% seems much impractical for real applications.

(7) Some terms are not well defined. For example, in section 3, AIR is not defined. AIR stands for Arithmetic Intermediate Representation in the context of zero-knowledge proof.

(8) Verification time is longer than the inference time. In Table 1, it takes 3.23s for a user to verify the correctness of the model with 79% accuracy, which is longer than the inference time. In the field of verifiable computation, the client’s verification effort is expected to be lower than performing the computation locally [g].

[a]zcash. halo2, 2022. URL https://zcash.github.io/halo2/.

[b] Ariel Gabizon, Zachary J Williamson, and Oana Ciobotaru. 2019. Plonk: Permutations over lagrange-bases for oecumenical noninteractive arguments of knowledge. Cryptology ePrint Archive (2019).

[c] Benoit Jacob, Skirmantas Kligys, Bo Chen, Menglong Zhu, Matthew Tang, Andrew Howard, Hartwig Adam, and Dmitry Kalenichenko. 2017. Quantization and Training of Neural Networks for Efficient Integer-Arithmetic-Only Inference. arXiv:1712.05877 [cs.LG]
[d] LEE, S., KO, H., KIM, J., AND OH, H. vcnn: Verifiable convolutional neural network based on zk-snarks. Cryptology ePrint Archive (2020).

[e] WENG, J., WENG, J., TANG, G., YANG, A., LI, M., AND LIU, J.-N. pvcnn: Privacy-preserving and verifiable convolutional neural network testing. IEEE Transactions on Information Forensics and Security 18 (2023), 2218–2233.

[f] LIU, T., XIE, X., AND ZHANG, Y. Zkcnn: Zero knowledge proofs for convolutional neural network predictions and accuracy. In Proceedings of the 2021 ACM SIGSAC Conference on Computer and Communications Security (2021), pp. 2968–2985.

[g] GHODSI, Z., GU, T., AND GARG, S. Safetynets: Verifiable execution of deep neural networks on an untrusted cloud. Advances in Neural Information Processing Systems 30 (2017).

[h] GROTH, J. On the size of pairing-based non-interactive arguments. In Advances in Cryptology–EUROCRYPT 2016: 35th Annual International Conference on the Theory and Applications of Cryptographic Techniques, Vienna, Austria, May 8-12, 2016, Proceedings, Part II 35 (2016), Springer, pp. 305–326.

**Questions:**

(1) Could you demonstrate how SoftMax is avoided in a detailed way, as is mentioned in section 4?

(2) Does the proving efficiency result from the efficiency of halo2, simplicity of MobileNet v2, or optimization proposed in this paper?

(3) Why does the verification time change from 10.27s to 3.23s, proving time from 2457.5s to 850.7s? Could you explain the optimization method behind this improvement?

(4) Can other verified machine learning methods also address the issue in the application scenarios mentioned section 5?

The reviewer would consider increasing the score after checking the authors' responses.

**Details Of Ethics Concerns:**

-

---

### Official Review · Reviewer_mW1C · 2023-11-01

**Soundness:** 3 good
**Presentation:** 3 good
**Contribution:** 2 fair
**Rating:** 5
**Confidence:** 3

**Summary:**

The paper introduces novel techniques to improve the efficiency and scalability of trustless verification for ML inference, even when dealing with complex datasets like ImageNet. The authors have leveraged a lookup table-driven approach to handle non-linear computations. The research addresses a pressing issue in ML verification using ZKP, particularly the challenge of scalability with complex datasets, making it relevant to the community.

**Strengths:**

1. The proposed method does not need interaction between the prover and the verifier, effectively reducing the need for communication overhead, typically associated with interactive protocols such as OT.


2.  Results are shown on the ImageNet dataset.


3. Methods are very well presented and easy to follow for a broader audience.

**Weaknesses:**

$\bullet$ **Scalability of the look-up-table-based approach:** While the authors claim scalability to ImageNet, they do not report top-1 accuracy. Furthermore, to better evaluate the effectiveness of the proposed method, it is essential to examine its performance on less complex datasets like CIFAR-10/100. In particular, the *major concern with the look-up-table-based approach lies in its scalability* from the viewpoint of the predictive performance of deeper models (such as ResNet52 and ResNet101)  on complex datasets, which is not shown in the paper.  Experimental results only with MobileNetV2 (and top-5 accuracy) are not good enough for claiming the scalability on imagenet.

$\bullet$ **Space complexity of took-up table:** The memory/storage overhead of the look-up table is not shown in the paper. Consequently, it remains unclear how the proposed non-interactive protocols trade off communication bandwidth (which is a crucial factor in interactive protocols such as OT) against memory usage. Additionally, it is not specified whether the verifier or prover is responsible for storing the lookup table (and whether it is feasible to have such memory capacity on the verifier/prover side).


$\bullet$ **Timing breakdown for linear and non-linear operations:** Given that non-linear computations are a significant bottleneck in trustless verification of machine learning inference using Zero-Knowledge Proofs, it would be insightful to include a timing breakdown (especially the proving and verification time, as shown in Table 1). Such an analysis could highlight the advantages of employing a look-up table-based method.

$\bullet$ The reuse of sub-circuits is not clearly explained in Section 4.

$\bullet$ **Minor issues with the writing and relevant literature:**

1. The same reference, "Safetynets: Verifiable execution of deep neural networks on an untrusted cloud" by Ghodsi et al., appears twice in the References section.

 2. The statements made in the related work section, claiming that Homomorphic Encryption (HE) is impractical and has only been deployed on toy datasets, are inaccurate. For instance, the reference "Cheetah: Lean and fast secure two-party deep neural network inference" by Huang et al. in USENIX Security 2022 demonstrates the practicality of HE in real-world deep neural network inference scenarios.

**Questions:**

1. Is there any particular reason for the selection of MobileNetV2?

See other points of weakness.

---

### Meta-Review · Area_Chair_nePR · 2023-12-05

**Metareview:**

This paper studies the problem of scaling up ZKP for DNN. Reviewers have many concerns about the paper while the authors did not provide a rebuttal to respond to those questions. In particular, reviewers are concerned about the novelty and writing of the paper. I would suggest the authors revising their paper accordingly and prepare for a next version for a future submission.

**Justification For Why Not Higher Score:**

The reviewers did not provide any rebuttal. So no reviewer wants to change their initial opinions, which are negative. Therefore, the paper is a clear rejection.

**Justification For Why Not Lower Score:**

N/A

---

### Decision · Program_Chairs · 2024-01-16

Reject